# The Effect of Cosmetic Ingredients of Phenol Type on Immediate Pigment Darkening and Their (Photo)Protective Action in Association with Melanin Pigmentation: A Model In Vitro Study

**Sara Viggiano** [1], **Lucia Panzella** [1] , **Maria Reichenbach** [2] , **Joachim Hans** [2,*] and **Alessandra Napolitano** [1,*]

1   Department of Chemical Sciences, University of Naples "Federico II", Via Cintia 4, I-80126 Naples, Italy
2   Cell & Molecular Biology, Global Innovation Cosmetic Ingredients R&D, Symrise AG, Muehlenfeldstr. 1, 37603 Holzminden, Germany
*   Correspondence: alesnapo@unina.it (J.H.); joachim.hans@symrise.com (A.N.)

**Abstract:** Immediate pigment darkening, the first response of skin to solar exposure leading to undesired irregular pigmentation and dark spots, is the rapid onset of melanin pigmentation resulting from oxidation of the melanogenic indoles, namely 5,6-dihydroxyindole (DHI) and 5,6-dihydroxyindole-2-carboxylic acid (DHICA) available in epidermal melanocytes. The search for effective sunscreen formulations is nowadays focused on UVA/B filters and additional ingredients that may scavenge the reactive oxygen species generated in these processes. In this work the effects of phenolic cosmetic ingredients (CIs), paradol-6, a ginger $CO_2$ extract, and phenylethyl resorcinol on photosensitized DHI and DHICA oxidation were investigated showing a decrease of their consumption and melanin formation (25–30% decrease with phenylethyl resorcinol). The photoprotective role of CIs was also evaluated in model systems. Paradol-6 and ginger $CO_2$ extract can halve linoleic acid peroxidation in the riboflavin-sensitized reaction, while dienes generation reduction (30% of control) was observed in the Rose-Bengal-sensitized photooxidation with paradol-6. The presence of DHI/DHICA melanin exerted a synergistic effect. The decay of thymine free or as a DNA base was almost completely inhibited by CIs. These results open new perspectives in the design of skin care formulations for ameliorating skin spots and contrasting ageing processes associated with sun exposure.

**Keywords:** UV-radiation; hyperpigmentation; immediate pigment darkening; melanin; gingerols; resorcinols; melanogenesis inhibition; 5,6-dihydroxyindole melanin precursors

## 1. Introduction

The pigmentary response of human skin to broadband UVA radiation (320–400 nm), causing several skin disorders, including melasma, freckles, age spots and other hyperpigmentation syndromes [1], occurs in three distinct phases. The first phase includes immediate pigment darkening (IPD) [2] represented by the production of dark pigments, melanins, from the oxidation of melanogenic indoles available in epidermal melanocytes [3,4]. Next, the second intermediate phase termed persistent pigment darkening (PPD)) [5] leads to the third phase of neomelanogenesis, or delayed tanning (DT).

The role of melanins in photoexposed skin is still controversial. Evidence that has been accumulating over the years has challenged the traditional view that melanins are efficient photoprotective agents sparing fundamental cellular constituents, particularly DNA and membrane lipid, from damages associated with solar radiation. Studies in vivo [6] and in vitro [7] seem to indicate that these pigments may act as photosensitizers on account of their significant absorption in the UV and visible region contributing to the production of reactive oxygen species (ROS) such as superoxide anion, singlet oxygen, hydroxyl radical and hydrogen peroxide capable of damaging DNA and other cellular structures [8,9].

Other evidence stemming from studies based on electron paramagnetic resonance (EPR) techniques would indicate that synthetic melanins are able to act as scavengers of the ROS they themselves generate on irradiation [10].

Based on these considerations, the search for effective sunscreen formulations is nowadays focused not only on the use of UVA/B filters to reduce the effects of carcinogenic and photodamaging solar UV radiation, but also on the development of additional ingredients that may scavenge the ROS [11]. Moreover, inhibitors/modulators of melanogenesis have been increasingly applied in skin care products for the treatment or prevention of abnormal hyperpigmentation [12]. The mechanism by which ROS scavenging may result in the inhibition of melanogenesis has been addressed in some studies. Raising levels of ROS may activate tyrosinase by mobilizing alpha-MSH in the epidermis, thus stimulating melanin synthesis in the melanocytes. Antioxidants can control the level of ROS and additionally can facilitate the dissociation of Nrf2, a step of the Keap1-Nrf2/ARE pathway that is activated at high ROS levels [13].

Among the most effective tyrosinase inhibitors, many natural phenolic compounds have been described that have also been demonstrated to exert an antioxidant activity [14–17].

On this basis, this work was directed towards the investigation of the effects of cosmetic ingredients (CIs) featuring a phenol moiety on the photooxidation of the indole melanin precursors 5,6-dihydroxyindole (DHI) and 5,6-dihydroxyindole-2-carboxylic acid (DHICA) taken as a model of the IPD process to obtain a preliminary assessment of the efficacy of these ingredients in the prevention of skin hyperpigmentation. In particular, gingerol-based Cis, namely paradol-6 and a ginger $CO_2$ extract containing [6]-gingerol as the main component, were evaluated together with a resorcinol derivative, phenylethyl resorcinol (Figure 1).

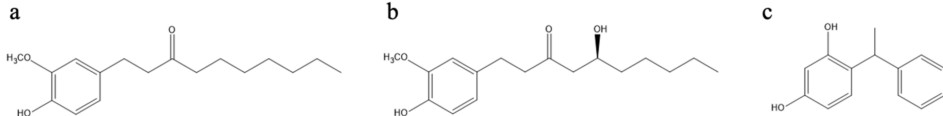

**Figure 1.** Structure of: (**a**) paradol-6; (**b**) main component of the ginger $CO_2$ extract; and (**c**) phenylethyl resorcinol.

Indeed, ginger is well known for its nutraceutical value, which can be ascribed to a variety of bioactive compounds including gingerols [18], the major pungent compounds, renowned for a variety of biological activities, ranging from anticancer, antioxidant, antimicrobial, anti-inflammatory and anti-allergic to various central nervous system activities.

On the other hand, resorcinol derivatives have been described as effective inhibitors of the initial stages of the melanin pigmentation pathway by trapping the tyrosinase generated dopaquinone [19], or affecting the signaling pathway that leads to tyrosinase production [20,21].

Yet, the possibility that these compounds may affect the IPD by interfering with melanogenic indole oxidation and associated ROS production has never been considered. The photoprotective role of these compounds against the damages induced by solar light was also evaluated by the use of modeling systems of lipid peroxidation and DNA damage. Additionally, the possibility that melanin pigmentation may affect the actions of CIs by exerting a synergic effect was also addressed by the use of model synthetic pigments.

## 2. Materials and Methods

### 2.1. Material

DHI and DHICA were synthetized using protocols previously developed [22,23].

CIs paradol-6, ginger $CO_2$ extract, phenylethyl resorcinol and 2-benzylaminobenzoic acid were provided by Symrise AG (Holzminden, Germany). Ginger $CO_2$ extract contains (6)-gingerol as the main component and the molecular weight of this compound was used to calculate the molar concentration of this CI throughout this study. DHI and DHICA

melanin were prepared by tyrosinase catalyzed oxidation of the indoles, as previously described [23].

Linoleic acid, liposomes of phosphatidylcholine from soybean containing 13% palmitic (C16:0), 4% stearic (C18:0), 10% oleic (C18:1), 64% linoleic (C18:2), and 6% linolenic (C18:3), thymine and calf thymus DNA, riboflavin and protoporphyrin IX were purchased from Merck and used as obtained.

### 2.2. Methods

Photoirradiation was carried out using a solar simulator (Thermo Oriel66902) at 60 watts or a multirays "merry-go-round" photoreactor (Helios Italquartz, Cambiago, Italy) equipped with lamps at 254, 310 and 366 nm.

UV-vis spectrophotometric analyses were performed on a Jasco V-730 spectrophotometer. HPLC analysis were carried out on an instrument equipped with an Agilent G1314A UV-vis detector using a $25 \times 0.46$ cm octadecylsilane column, 5-$\mu$m particle size. Elution conditions: 1% formic acid/acetonitrile 75:25 $v/v$ with detection set at 280 nm (DHI and DHICA analysis); 10 mM phosphate buffer (pH 6.8) with detection wavelength set at 220 nm (thymine and thymine dimers analysis).

### 2.3. Photosensitized Oxidation of DHI and DHICA

The photoirradiation was carried out using a solar simulator on solutions of DHI or DHICA at 50 $\mu$M concentration in 0.1 M phosphate buffer at pH 7.0. The solutions were exposed to the solar simulator in a Petri dish that has been covered with a quartz slide and periodically analyzed by UV-vis spectrophotometry or HPLC without further dilution or work up. The photosensitizer (PS) (riboflavin or protoporphyrin IX) was added to the solution of the indoles at 1 $\mu$M final concentration. When required, CIs or 2-benzylaminobenzoic were added to the mixture at the same molar concentration of the indoles. Control experiments were run under the same conditions with one of the ingredients of the mixture (PS, or the indole) lacking.

### 2.4. Lipid Photooxidation

Linoleic acid at 2.5 mM concentration in 0.1 M phosphate buffer at pH 7.0 containing 0.1 M sodium dodecyl sulphate (SDS) in the presence of the appropriate photosensitizer at different doses was exposed to the solar simulator radiation in a Petri dish that was covered with a quartz slide to avoid evaporation of the solvent. The mixture was periodically analyzed by UV-vis spectrophotometry without further dilution or work-up to improve the reproducibility of the experiments. When required, CIs, DHI melanin (DHI-mel) or DHICA melanin (DHICA-mel) were also included. The melanins were added as a suspension in water after careful homogenization [24].

In other experiments, 250 $\mu$M suspension of liposomes in 0.1 M phosphate buffer at pH 7.0 were exposed to the solar simulator in the presence of the photosensitizer, with or without the CIs.

### 2.5. Ferrous Oxidation Xylenol Orange (FOX) Assay

Linoleic acid solution at 2.5 mM (250 $\mu$L) containing riboflavin at 20 $\mu$M were withdrawn before and after exposure to the solar simulator and were mixed with 2 mL of reagent mixture containing 0.11 mM xylenol orange, 0.25 mM ammonium iron (II) sulfate hexahydrate and 25 mM $H_2SO_4$ in methanol containing 3.88 mM butylhydroxyanisole, BHA. After 30 min incubation in the dark at room temperature, absorbance at 560 nm was measured. The same experiment was repeated in the presence of paradol-6 (20 $\mu$M) and/or in the presence of DHI-mel (0.05 mg/mL).

### 2.6. Photoinduced Decay of Free or DNA Thymine

The photoirradiation was carried out using a multirays "merry-go-round" photoreactor on a 2 mM thymine solution in 0.1 M phosphate buffer (pH 7.0). The solutions were

exposed to the photoreactor in quartz tubes and were periodically analyzed by HPLC after dilution to 80 µM concentration. 10 mM phosphate buffer (pH 6.8) was used as the eluant.

### 2.7. Photoinduced DNA damage

A solution of DNA was prepared dissolving 25 mg of calf thymus DNA in 4 mM Tris–HCl buffer solution (pH 7.4) containing 0.4 mM EDTA (to prevent nucleases from degrading DNA). According to manufacturer instructions, to reduce shearing of the DNA, no sonication or stirring was used. Complete dissolution requires ca. 24 h. 500 µL of the solution was irradiated in a quartz tube in the presence or in the absence of each CI (50 µM) for 3 h. 30 µL of the irradiated solution and 1 mL trifluoroacetic acid were combined in a glass vial in an argon atmosphere. The mixture was heated under stirring for 1 h at 160 °C, cooled and dried under vacuum. The residue was dissolved in water (200 µL) and left at room temperature overnight prior to injection in the HPLC system. 10 mM phosphate buffer (pH 6.8) was used as the eluant, on an octadecylsilane column as above with detection wavelength set at 220 nm.

## 3. Results and Discussion

### 3.1. In Vitro Model of IPD: Photosensitized Oxidation of Melanogenic Indoles and Effects of the CIs

In the initial experiments the course of the photooxidation reaction of DHI and DHICA in phosphate buffer at pH 7.0 was carried out using a solar simulator. The reaction course was followed by UV-vis spectrophotometry by periodical measurement of the spectrum over the wavelength range 200–800 nm over the first 30 min. Yet, under these conditions both DHI and DHICA absorption spectra in the UV region remain unchanged. For comparison the aerobic oxidation of the two indoles was monitored over the same time period, showing no significant consumption (data not shown).

The modest or null consumption of the substrate following UV-vis irradiation prompted us to consider the use of photosensitizers (PS). Two were selected, namely protoporphyrin IX (pPIX) and riboflavin (RF) that are known to act with different mechanisms. Riboflavin is capable of generating either superoxide/hydrogen peroxide or singlet oxygen and is generally ranked as type I/II photosensitizer while protoporphyrin IX is a type II photosensitizer producing only singlet oxygen [25].

The PS selected were natural or mimicking those occurring in the skin. Their amount was optimized in different experiments, and either proved able to induce the conversion of DHI or DHICA to melanin over 30 min, as apparent from the increase of the absorbance in the visible region due to the scattering of the insoluble pigment generated (data not shown).

Optimization of the reaction time and indole concentration was then addressed. A reaction time of 1 h was eventually chosen since this afforded more reproducible results; on the other hand, the use of concentrations (e.g., 50 µM) that allowed for direct measurements without dilution also contributed to improve the reproducibility of the experiments. In addition, control experiments showed that both PS were not transformed following photoirradiation over the same time, nor did they interact with the indoles under aerobic conditions in the absence of irradiation.

The effects of the selected CIs on the photosensitized oxidation of DHI and DHICA were then investigated using a 1:1 molar ratio. Melanin formation in the DHI photoirradiation mixture was estimated by analyzing the changes in the absorbance at 600 nm selected in the visible region to avoid interference of the absorption of the CI. Note that for each experiment with a different CI, a separate blank experiment in the absence of CI was run to allow for a more reliable comparison.

The results shown in Figure 2 indicate that in the case of paradol-6, ginger $CO_2$ extract, and phenylethyl resorcinol, a slight though non-statistically significant decrease of melanin formation from DHI photosensitized irradiation was observed.

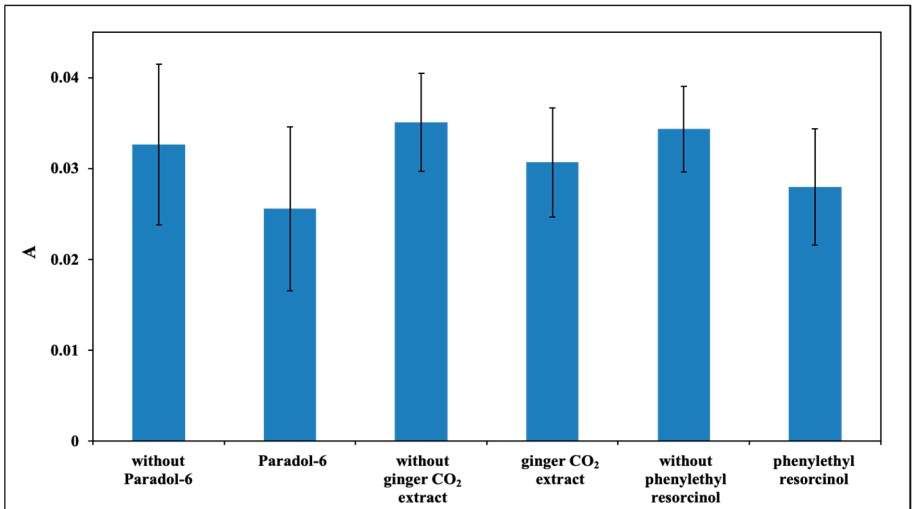

**Figure 2.** Absorbance at 600 nm of DHI photoirradiation mixture containing 1 μM RF with or without selected CI (paradol-6, ginger $CO_2$ extract, phenylethyl resorcinol) after 1 h.

Control experiments were run using the selected CI and RF in the absence of the indoles or the CI and the indoles in the absence of RF. Using paradol-6 and phenylethyl resorcinol as CIs, slight changes in the absorption spectrum were observed after irradiation in the presence of the PS. On the contrary, in the absence of the PS no appreciable decay was observed, indicating that a photosensitizer is needed to start the photooxidation using the solar simulator (data not shown).

To evaluate the kinetic of DHI and DHICA decay in the photooxidation mixtures we moved to HPLC analysis since direct analysis of the spectra in the UVA/B region proved very troublesome due to overlapping of the chromophores of the ingredients of the reaction mixture.

In the presence of RF as PS, the irradiation led to a high consumption of DHI, around 80% (Figure 3a). The addition of equimolar CIs reduced the consumption of the indole. In particular, the most intense effect was observed with phenylethyl resorcinol, which reduced the consumption of the indole to ca. 60% after 1 h.

The same set of analyses were carried out with DHICA (Figure 3b). Also in this case, in the presence of RF, the consumption of the indole at 1 h irradiation time was around 80%. The addition of equimolar CIs effectively reduced this decay; in particular, best results were obtained using paradol-6 and phenylethyl resorcinol leading to an approximately 50% consumption of the indole.

The same experiments were run using protoporphyrin IX as PS. Yet, in the presence of DHI the effects were far different from those observed in the RF-photosensitized reaction. Indeed, while no ameliorative effects were observed with paradol-6 and ginger $CO_2$ extract, even higher indole consumption was observed with phenylethyl resorcinol (Figure 4). Similar results were obtained for DHICA (not shown).

Mechanistic Insights

The difference of the effects of the CIs on the indole photooxidation induced by the two PS selected would suggest that the CIs tested may act as scavengers only of the species generated by RF (superoxide and hydrogen peroxide) but not of the singlet oxygen generated by pPIX.

A paradigmatic example is represented by phenylethyl resorcinol, that showed opposite effects on DHI melanisation with the two different PS investigated. Indeed, as mentioned above, phenylethyl resorcinol reduced the consumption of DHI using RF as PS, while the opposite effect was observed in the presence of pPIX (Figures 3a and 4).

To confirm this hypothesis, in further experiments the irradiation under solar simulator light was carried out using 2-benzylaminobenzoic acid, belonging to the amino benzoic acid class of compounds that are known to act as selective scavengers of singlet oxygen [26,27].

The effects of this CI on the photosensitized oxidation of DHI and DHICA were investigated using a 1:1 molar ratio over 1 h reaction time, as in the previous experiments. The RF-sensitized irradiation carried out without the CI led to a consumption of more than 80% for DHI (Figure 5a) and of approximately 75% for DHICA (Figure 5b), while the presence of equimolar 2-benzylaminobenzoic acid led to a comparable or even higher decay of both DHI and DHICA at 1 h (Figure 5a,b).

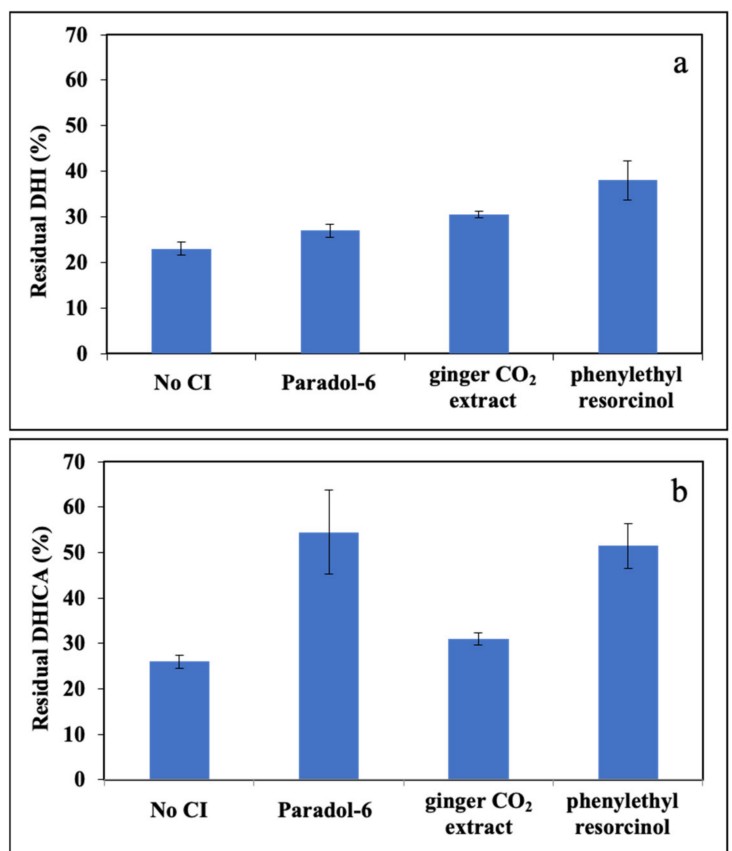

**Figure 3.** Residual concentration of: (**a**) DHI; or (**b**) DHICA after 1 h photoirradiation in the presence of 1 μM RF and selected CIs (paradol-6, ginger $CO_2$ extract and phenylethyl resorcinol).

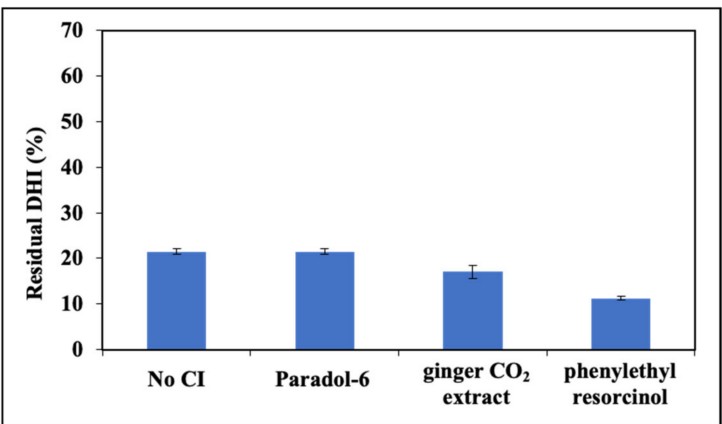

**Figure 4.** Residual concentration of DHI after 1 h photoirradiation in the presence of 1 μM pPIX and selected CIs (paradol-6, ginger $CO_2$ extract and phenylethyl resorcinol).

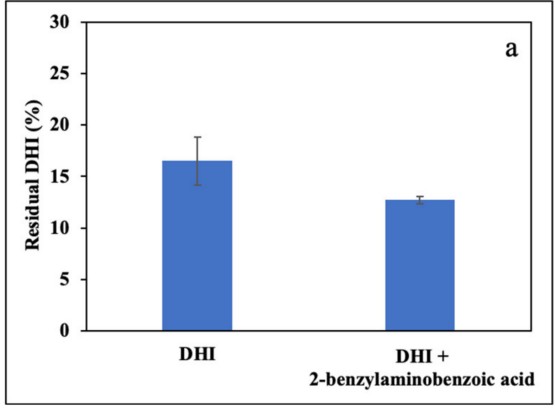
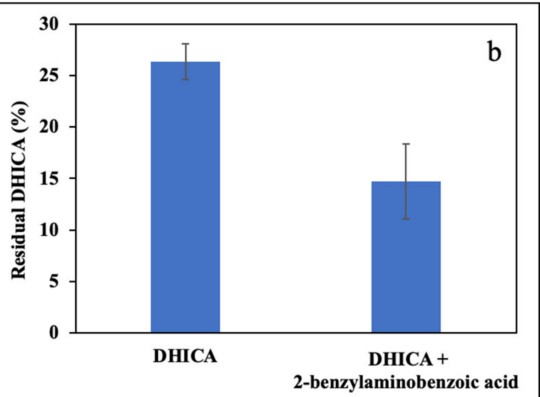

**Figure 5.** Residual concentration of: (**a**) DHI; or (**b**) DHICA after 1 h photooxidation in the presence of 1 µM RF and with or without equimolar 2-benzylaminobenzoic acid. Data are expressed as mean ± SD of three independent experiments, each performed in duplicate.

Different results were obtained using pPIX as PS. While the consumption of the monomer was almost complete both in the case of DHI and DHICA, using 2-benzylaminobenzoic acid the consumption of either indole was significantly reduced (Figure 6a,b).

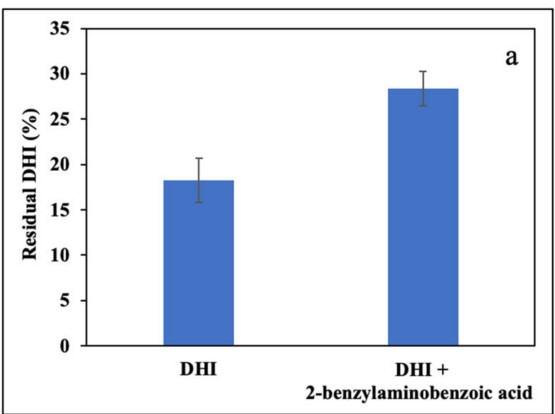
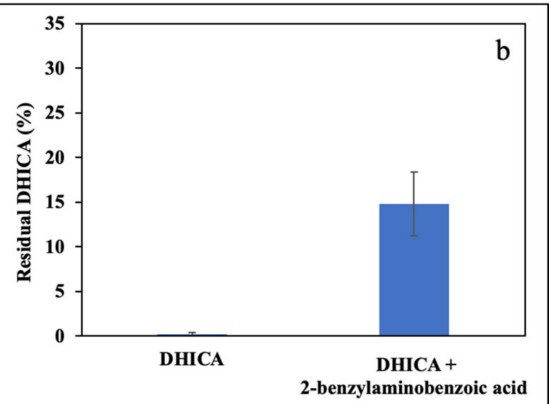

**Figure 6.** Residual concentration of: (**a**) DHI or (**b**) DHICA after 1 h photooxidation in the presence of 5 µM pPIX and equimolar 2-benzylaminobenzoic acid. Data are expressed as mean ± SD of three independent experiments, each performed in duplicate.

Therefore, 2-benzylaminobenzoic acid proved able to protect the indoles from photooxidation sensitized by pPIX but not RF, confirming the above proposed hypothesis.

*3.2. Photosensitized Lipid Peroxidation*

The actual role of melanin in model biological systems in the presence of other species capable of scavenging or quenching superoxide or singlet oxygen is still to be defined. Thus, it seemed interesting to evaluate the potential of selected CIs of phenol type in protecting biological targets from reactive oxygen species produced on irradiation or following oxidative stimuli and to assess the effects of melanins.

A convenient model for studying oxidative damages to lipid membranes is represented by soybean phosphatidylcholine liposomes at high content (around 70%) of polyunsaturated fatty acids that can undergo lipid peroxidation [28]. Since polyunsaturated fatty acids (PUFA) are the main target of lipid peroxidation, linoleic acid was initially selected as easily accessible substrate. However, PUFA do not absorb the radiation in the emission region of the solar simulator, so a photosensitizer is needed.

Photosensitized oxidation can take place through two different pathways [25]:

- type I mechanism: the generation of radicals via electron transfer or hydrogen abstraction;

- type II mechanism: energy transfer from the triplet state of the photosensitizer to the molecular oxygen to form $^1O_2$.

In the case of PUFA, both mechanisms are possible [29], so riboflavin (a type I/II photosensitizer), was used to induce lipid peroxidation.

Lipid peroxidation of PUFA can be followed by monitoring the development of primary oxidation products, namely, conjugated dienes (conjugated hydroperoxides), and trienes (carbonyls conjugated to dienes), absorbing at 234 and 270 nm, respectively.

However, the great contribution of RF absorption at 270 nm made it difficult to follow the generation of trienes, so attention was focused on formation of dienes.

Optimization of the reaction conditions (time and photosensitizers concentration) was initially pursued. A reaction time of 15 min was eventually chosen since extending exposure time did not induce an increase in absorbance at 234 nm but led, on the contrary, to a decrease in the concentration of conjugated dienes, suggesting further reactions leading to secondary products, as documented in several studies [30,31].

In the case of RF, an optimal concentration of 20 μM was chosen that warranted appreciable diene formation over the selected reaction time. Experiments were carried out also using pPIX and Rose Bengal, as PS. The first one was used at increasing concentrations; however, in no case could an appreciable peroxidation of linoleic acid be induced. Therefore, we moved to Rose Bengal (RB), a type II photosensitizer [32], that proved able to induce a significant formation of conjugated dienes at 20 μM over 15 min.

Control experiments on linoleic acid emulsions were carried out under the same conditions described above with or without RF. Without irradiation no significant autooxidation and generation of conjugated dienes occurred over the reaction time selected. Furthermore, negligible changes in the absorption of RF were detected during the irradiation, indicating no significant photodegradation of this photosensitizer. Control experiments were also run for the RB-sensitized oxidation, showing no appreciable contribution of the PS at 234 nm also after irradiation.

3.2.1. Effects of CIs on the Photooxidation of Linoleic acid and Assessment of the Role of Melanins

The optimized conditions of the photosensitized oxidation of linoleic acid were used to evaluate the effects of the selected CIs. The variation of absorbance (ΔA(%)), as shown in Figure 7, represents the percent variation referred to the control, assuming lipid peroxidation in the absence of CIs ingredients as 100%.

Paradol-6 and ginger $CO_2$ extract are capable of halving the generation of dienes in the RF-sensitized reaction (Figure 7a,c). On the other hand, paradol-6 and phenylethyl resorcinol performed better with the RB-sensitized photooxidation, with a decrease of dienes generation to 30% of the control in the case of paradol-6 (Figure 7b,d).

In the case of paradol-6 and ginger $CO_2$ extract, dose–response curves for the RF-sensitized process were recorded (Figure S1). It appears that at 40 μM both of them reach their $IC_{50}$ dose. Phenylethyl resorcinol proved much less effective in the RF-sensitized reaction (not shown). On the other hand, in the RB-sensitized process (Figure S1), paradol-6 reaches its $IC_{50}$ dose at concentrations as low as 10 μM and phenylethyl resorcinol at 20 μM, whereas ginger $CO_2$ extract were less effective with $IC_{50}$ at around 40 μM.

In control experiments the contribution to the absorbance (ΔA(%)) at 234 nm using the CIs and RF in the absence of linoleic acid was evaluated and this correction was introduced in all further experiments. The contribution of CIs to the absorption following irradiation in the absence of the PUFA, but in the presence of RB was as well assessed.

To investigate the possible effects of melanins in the photosensitized oxidation of linoleic acid in the presence of CIs, synthetic pigments obtained by enzymatic oxidation of DHI and DHICA were used following several studies showing that such melanins are good models of natural pigments and moreover warrant a higher degree of structural homogeneity [33,34].

Control experiments showed that DHICA-mel (0.0125 mg/mL) and, to a larger extent DHI-mel (0.05 mg/mL), caused a small increase of lipid peroxidation (data not shown). A lower concentration of DHICA-mel with respect to DHI-mel was used in order to limit its absorption contribution at 234 nm.

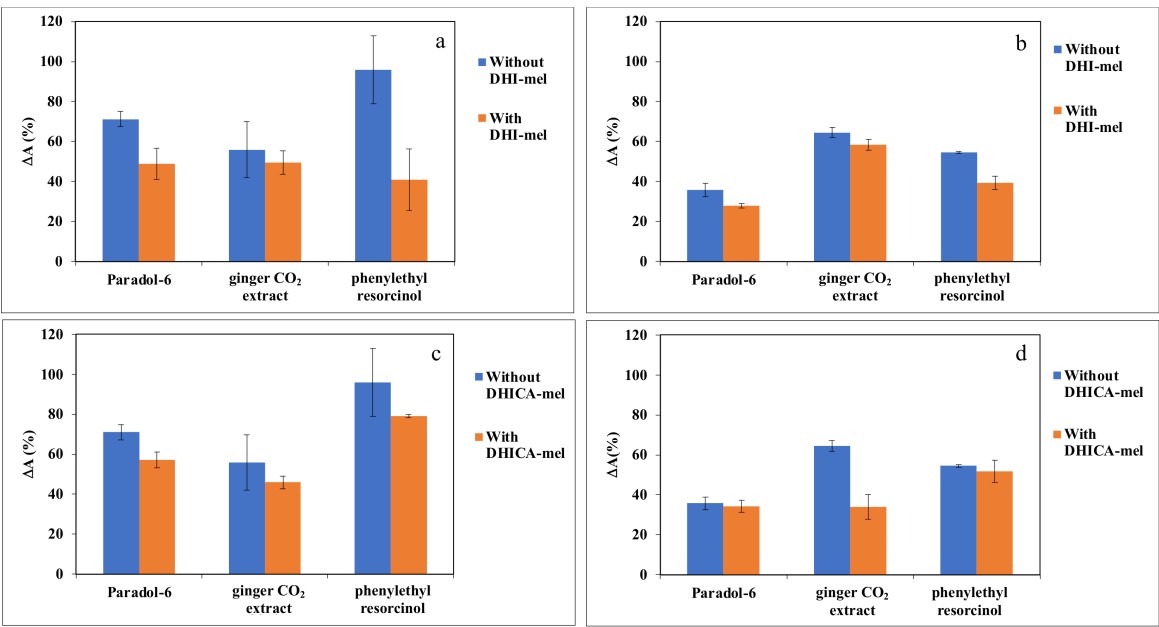

**Figure 7.** Changes of the absorbance at 234 nm of 2.5 mM linoleic acid oxidation mixture containing: 20 μM of RF (**a**,**c**); or 20 μM of RB (**b**,**d**), 20 μM of selected CI (paradol-6, ginger $CO_2$ extract and phenylethyl resorcinol) and 0.05 mg/mL DHI-mel (**a**,**b**); or 0.0125 mg/mL DHICA-mel (**c**,**d**). Data are expressed as mean ± SD of three independent experiments, each performed in duplicate.

In the case of the RF-sensitized reaction, the presence of DHI-mel together with paradol-6, phenylethyl resorcinol or ginger $CO_2$ extract reduced lipid peroxidation by more than 50%, while with DHICA-mel, all CIs can decrease the extent of damage albeit to varying degrees (Figure 7a,c).

The effects of the presence of melanins were significant also in the case of the RB-photosensitized linoleic acid oxidation (Figure 7b,d). In particular, only the presence of DHICA-mel proved able to lower the oxidation of some 40% (not shown) while, in the addition of paradol-6 or ginger $CO_2$ extract a decrease of approximately 70% was observed.

The dose–response curves were plotted for the RF-sensitized process using the CIs that gave the best results, paradol-6 and ginger $CO_2$ extract, in the presence of DHI-mel or DHICA-mel (Figure S2). Paradol-6 exhibited a marked dose–response with more than doubling of the effects on lipid peroxidation passing from 20 μM to 40 μM concentration, in the presence of both melanins.

In order to confirm that the effects observed were not altered by the correction introduced to take into account the contribution to the absorption at 234 nm of the control mixture (without linoleic acid), an alternative procedure was used for determination of hydroperoxides. The selected method was the ferrous oxidation xylenol orange (FOX) assay, a rapid and inexpensive method commonly used to detect hydroperoxides generated during the early stages of lipid peroxidation. Hydroperoxides oxidize ferrous to ferric iron and the resulting ferric ion binds to xylenol orange to produce a colored complex with a strong absorbance at 560 nm [35]. The assay run using paradol-6 as a representative CI, confirmed the obtained results (Figure S3).

### 3.2.2. Soybean Phosphatidylcholine Liposome Peroxidation under Solar Simulator Irradiation

Based on these results, the photoinduced lipid oxidative damage was investigated on a more complex model, namely, the liposomes of phosphatidylcholine from soybean (SoyPC). This is a generally used model because phosphatidylcholine is the most abundant phospholipid in eukaryotic cellular membrane, and because it contains high amounts of PUFAs.

Despite long exposure times and high concentrations of RF, it was not possible to induce lipid peroxidation by photoirradiation under the solar simulator. Actually, liposomes have been described as being able to stabilize RF [36].

Only an appreciable degree of peroxidation could be obtained using RB. None of the CIs tested were able to inhibit the oxidation induced by this photosensitizer. Taking in account that RF is able to act through either type I and type II mechanism, while RB can only act through the generation of singlet oxygen, it is likely that paradol-6 and ginger $CO_2$ extract can only control and/or scavenge radical oxygen-derived species.

Also in this case, to support this hypothesis, the effects of 2-aminobenylbenzoic acid were evaluated. Indeed, it proved able to inhibit liposomes peroxidation by ca. 20% and this effect was strongly enhanced in the presence of DHI-mel, whereas melanin alone did not have any effect (Figure S4).

### 3.3. Photoinduced Decay of Thymine by UV Irradiation

Thymine is one of the nitrogenous bases most sensitive to UV-induced damage, leading to different photoproducts including cyclobutane dimers. For this reason, this compound was preliminarily chosen as a model system to evaluate the potential protective action of the CIs against UV-induced damages of pyrimidine bases in DNA.

The assessment of cyclopyrimidine dimers formation in DNA produced by UV irradiation can be finely mimicked by following the formation of the thymine-thymine dimer or determination of its levels in the DNA hydrolysate after irradiation [37].

A straightforward method was adopted to monitor the process based on HPLC analysis of thymine solutions subjected to irradiation. A standard solution of dimers was prepared using a modified procedure reported in the literature [37] in which a 2 mM solution of thymine was frozen and photoirradiated at 254 nm for 10 min. Thymine consumption (T in Figure 9, $R_t$ = 15 min) and photoproducts generation was followed by RP-HPLC with UV-detection fixed at 220 nm in order to obtain the highest sensitivity for both thymine monomer and dimers. Peaks at $R_t$ 10 and 14 min (T-T in Figure 8) were attributed to thymine dimers after LC-MS analysis ([M+H]$^+$ = 253 $m/z$).

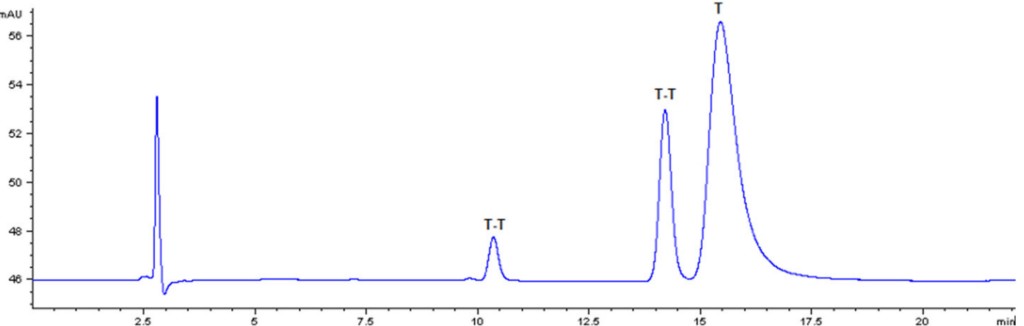

**Figure 8.** Elution profile of a thymine (T) solution at 80 μM photoirradiated at 254 nm for 10 min. Detection wavelength 220 nm. T-T: thymine dimers.

Experimental conditions were then optimized. Three-hour exposure to a UV lamp at 254 induced ca. 50% of thymine consumption. Irradiation was also run in an argon atmosphere or in the presence of a reducing agent (sodium bisulfite). Under these conditions, residual thymine was higher (77% under argon atmosphere, 93% in the presence of sodium bisulfite, vs. 51% under the standard conditions) suggesting that, in addition to dimer

formation, other reaction routes involving oxidation of thymine are operative. Control experiments indicated that no significant consumption of thymine was observed over the three-hour exposure to air without irradiation. LC-MS analysis provided evidence of the presence of other oxidation products, such as 5-formyluracil ([M+H]$^+$ = 141 *m/z*) in the mixture photoirradiated in air.

### 3.3.1. Effects of CIs on the Photoinduced Thymine Decay and Assessment of the Role of Melanins

The effects of CIs on thymine photodimerization were then examined. Due to overlap of the CI elution, peaks with thymine dimers in the elutographic conditions developed, and it was eventually chosen to monitor only the consumption of thymine after irradiation. Figure 9 shows residual thymine in the presence of all CIs. It appears that in the presence of paradol-6, ginger CO$_2$ extract and phenylethyl resorcinol, the decay of thymine is almost completely inhibited. Dose–response curves confirmed this trend (Figure S5).

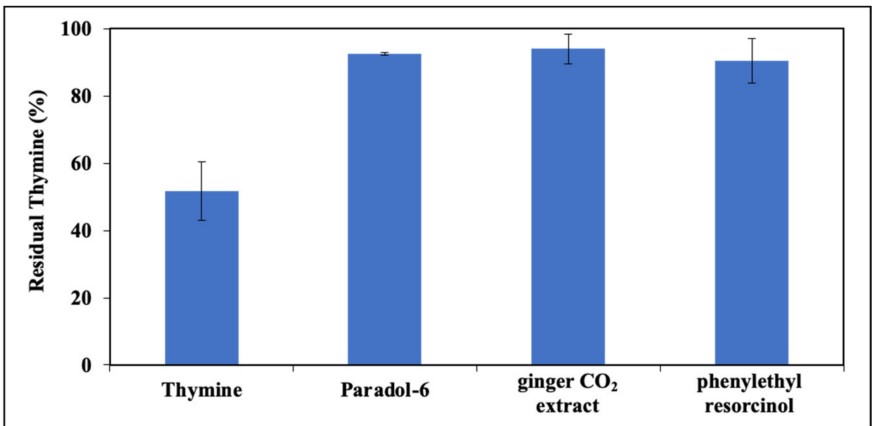

**Figure 9.** Residual thymine expressed as percent compared to control after photoirradiation at 254 nm at 3 h. Data are expressed as mean ± SD of three independent experiments, each performed in duplicate.

The effect of DHI-mel and DHICA-mel was then evaluated. Preliminary experiments showed that a comparable protection is obtained using DHI-mel at 0.05 mg/mL concentration (73%) or DHICA-mel at 0.004 mg/mL (72%). This result is in line with previous observations indicating that DHICA-mel is a better antioxidant than DHI-mel [33].

The effect of the combination of CIs (paradol-6 or ginger CO$_2$ extract) and melanins was evaluated showing only a small synergic effect (Figure S6).

In a final set of experiments, the effects of CIs on thymine photodegradation as induced under solar simulator light in the presence of RF was investigated. The most effective was paradol-6 and phenylethyl resorcinol with ca. 30% increase of residual thymine, while a modest effect (10%) was observed in the presence of ginger CO$_2$ extract (Figure S7).

### 3.3.2. Effect of CIs on Photoinduced DNA Damage

In further experiments, a previous methodology [37] was adapted to evaluate the UV-induced damage in deoxyribonucleic acid (DNA) based on thymine consumption, following photoirradiation at 254 nm (as described above).

Chromatographic analysis was carried out on hydrolysate of DNA following the irradiation and thymine was quantified against those of a DNA sample that had not been photoexposed. Chromatographic conditions were optimized to avoid interferences with other products generated by the hydrolysis of DNA. Moreover, the conditions of the hydrolysis were properly chosen to have good run-to-run reproducibility of the levels of thymine.

When CIs were included in the photoirradiation mixture, a general protective effect on the decay of DNA thymine bases was observed, with paradol-6 and phenylethyl resorcinol

proving to be the most active (Figure 10), in good agreement with the results of the experiments on free thymine.

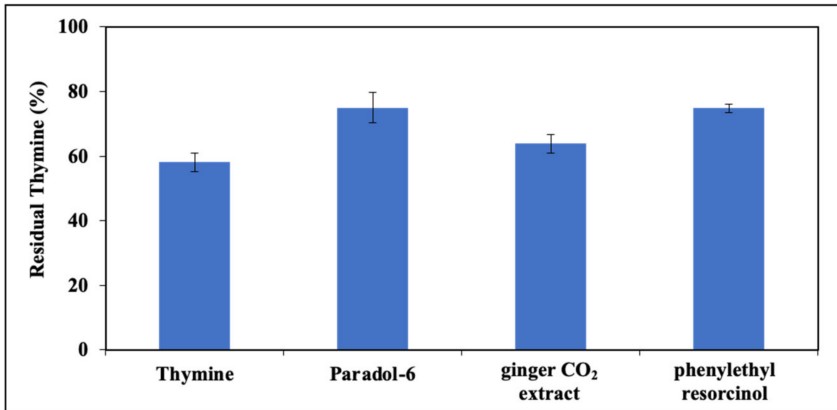

**Figure 10.** Residual thymine in DNA expressed as percent compared to control after UV exposure (3 h). Data are expressed as mean ± SD of three independent experiments, each performed in duplicate.

## 4. Conclusions

The possibility to expand the potential of chemical ingredients of skin-care products designed for ameliorating skin spots and contrasting ageing processes including those associated with sun exposure is a very interesting perspective in dermo-cosmetic research. In this regard, the IPD process, which should warrant immediate protection to photoexposed skin leading to production of melanins before the activation of the complex enzymatic machinery of the biosynthesis, may result in undesired inhomogeneous pigmentation. In this work taking the oxidation of the ultimate biosynthetic precursors of melanins by solar simulator irradiation in the presence of type I and type II photosensitizers (riboflavin and protoporphyrin IX) as a model of the IPD process, we investigated the effects of selected chemical ingredients belonging to the class of gingerols and resorcinols. Indeed, the difference observed in the photoprotective action exerted by the CIs indicated that these may act as scavengers of the species generated by riboflavin (superoxide and hydrogen peroxide), but not of the singlet oxygen generated by protoporphyrin IX. Considering that the most effective CIs were those exhibiting an intense absorption in the UVB with partial covering of the UVA region, the possibility that a screening action in the UVA region sparing the indoles contributes to some extent to the observed effects that should be considered.

The photoprotective role of CIs against the damages induced by solar light was also evaluated by the use of model systems of lipid peroxidation and DNA damage. All the CIs were effective in inhibiting the photoinduced lipid peroxidation in a dose-dependent manner. The role of melanin pigmentation in these processes was also investigated by the use of model synthetic pigments obtained by biomimetic oxidation of the melanogenic indoles. Interestingly enough, a synergic effect was observed in the presence of DHI-mel with a strong dose dependence for paradol-6 in the RF-sensitized linoleic acid oxidation, while DHICA-mel proved more effective in the RB-induced lipid photooxidation.

The selected CIs were also very effective in preventing photodimerization of thymine either free or as a DNA base, in the UV- or RF-photosensitized solar simulator irradiation.

The main outcomes of this study are presented in a pictorial view in the scheme of Figure 11.

Though of interest, the results of this study await confirmation from further experiments using ex vivo or in vivo systems that could better simulate the biological processes that have only been mimicked in the present work. Moreover, the effects of melanin pigmentation in tissues can hardly be simulated by the use of finely suspended melanin pigments from the indole precursors.

Notwithstanding these limitations, the results obtained open new perspectives in the design of skin-care formulations showing differences in the mechanism of action of

commonly used gingerol type and resorcinol ingredients and expand the knowledge of their protective effects against solar-induced damages to cellular components. In addition, a hitherto underestimated role of melanin in reinforcing the protective action of these chemical ingredients was highlighted.

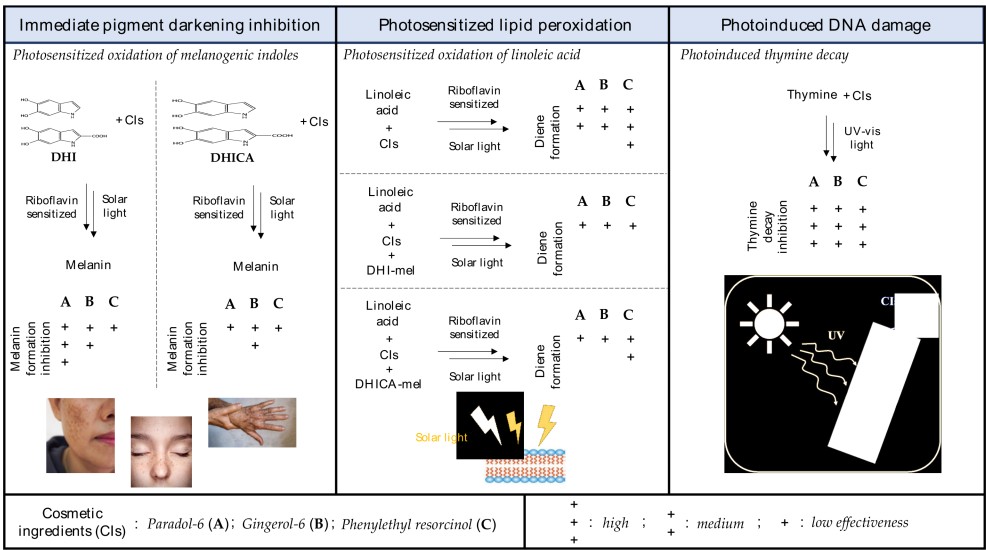

**Figure 11.** Summarizing scheme of the main outcomes of the study.

**Supplementary Materials:** The following supporting information can be downloaded at: https://www.mdpi.com/article/10.3390/cosmetics10010022/s1, Figure S1: dose–response curves of linoleic acid photooxidation for paradol-6 and ginger $CO_2$ extract in the RF-sensitized reaction (a, b) and for paradol-6, ginger $CO_2$ extract and phenylethyl resorcinol in the RB-sensitized process (c, d, e); Figure S2: dose–response curves of linoleic acid photooxidation for RF-sensitized process using paradol-6 or ginger $CO_2$ extract in the presence of 0.05 mg/mL DHI-mel (a, b) or 0.0125 mg/mL DHICA-mel (c, d); Figure S3: Comparative evaluation of lipid peroxidation by diene formation monitoring at 234 nm and hydroperoxide measurement by FOX assay; Figure S4: Effect of 2-aminobenylbenzoic acid (20 µM) on photoinduced peroxidation of a 250 µM suspension of liposomes containing 20 µM RB and/or 0.05 mg/mL DHI-mel; Figure S5: dose–response curves for paradol-6, ginger $CO_2$ extract and phenylethyl resorcinol in thymine photoirradiation.; Figure S6: Effect of paradol-6 and ginger $CO_2$ extract on thymine consumption by photoirradiation in the presence and in the absence of DHI- or DHICA-mel (0.05 and 0.004 mg/mL, respectively); Figure S7: Residual thymine expressed as percent compared to control after irradiation with the solar simulator in the presence of CIs at 50 µM and RF 50 µM over 30 min.

**Author Contributions:** Conceptualization, L.P. and A.N.; methodology, S.V. and M.R.; investigation, S.V.; writing—original draft preparation, A.N.; writing—review and editing, J.H. All authors have read and agreed to the published version of the manuscript.

**Funding:** This research was funded by Symrise AG project no 163477 and in part by 2017YJMPZN PRIN project.

**Institutional Review Board Statement:** Not applicable.

**Informed Consent Statement:** Not applicable.

**Data Availability Statement:** The data presented in this study are available in the Supplementary Material.

**Conflicts of Interest:** The authors declare no conflict of interest.

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
