# Peer review of "The Effect of Cosmetic Ingredients of Phenol Type on Immediate Pigment Darkening and Their (Photo)Protective Action in Association with Melanin Pigmentation: A Model In Vitro Study"

_cosmetics, doi:10.3390/cosmetics10010022_

Round 1
Reviewer 1 Report
This is an interesting manuscript on a well designed and written study. The references and figures are nicely presented and methods are well described and easily reproduced. I only have few comments:
1. what is a model study? is this a pilot? I am not familiar with this design from the title
2. add settings, results (numbers, p values) to the abstract
3. line 135 - never start a sentence with a number
4. please add limitation section
Reviewer 2 Report
This is a perfect paper on indoles and indoloquinones in melanogenesis, and their influence by some types of photosensitizers in the presence or absence of melanin, and/or DNA/Thymidine. I cannot find any drowbacks of the study. I would like to ask for two improvements (if possible):
1. As the process described and analyzed in the paper, including DHI or DHICA, photosensitizers, melanin, DNA and oxygene is multidimensional, one could do with a resultant, generalized scheme illustrating the whole process described in details in the experiments (supplementing the conclusions)
2. Melanins were investigated as carefully homogenized water suspensions (line 130). May it be affected in other way by other environments of the observed reactions?
